# Intersection of Redox Chemistry and Ubiquitylation: Post-Translational Modifications Required for Maintaining Cellular Homeostasis and Neuroprotection

**DOI:** 10.3390/cells10082121

**Published:** 2021-08-18

**Authors:** Emma I. Kane, Kelly L. Waters, Donald E. Spratt

**Affiliations:** Gustaf H. Carlson School of Chemistry and Biochemistry, Clark University, 950 Main St., Worcester, MA 01610, USA; emkane@clarku.edu (E.I.K.); kwaters@clarku.edu (K.L.W.)

**Keywords:** neurodegenerative disease, ubiquitin-proteasome system, redox chemistry, post-translational modifications, neuroprotection, CNS, mitochondrial metabolism, oxidative stress response

## Abstract

Neurodegeneration has been predominantly recognized as neuronal breakdown induced by the accumulation of aggregated and/or misfolded proteins and remains a preliminary factor in age-dependent disease. Recently, critical regulating molecular mechanisms and cellular pathways have been shown to induce neurodegeneration long before aggregate accumulation could occur. Although this opens the possibility of identifying biomarkers for early onset diagnosis, many of these pathways vary in their modes of dysfunction while presenting similar clinical phenotypes. With selectivity remaining difficult, it is promising that these neuroprotective pathways are regulated through the ubiquitin-proteasome system (UPS). This essential post-translational modification (PTM) involves the specific attachment of ubiquitin onto a substrate, specifically marking the ubiquitin-tagged protein for its intracellular fate based upon the site of attachment, the ubiquitin chain type built, and isopeptide linkages between different ubiquitin moieties. This review highlights both the direct and indirect impact ubiquitylation has in oxidative stress response and neuroprotection, and how irregularities in these intricate processes lead towards the onset of neurodegenerative disease (NDD).

## 1. Introduction

Neurodegenerative disease (NDD) research initiated approximately four decades ago, but recently, the focus has shifted to understanding both the neurochemistry aspect in disease development and neuroprotection. Proper central nervous system (CNS) development plays an essential role in neuroprotection, as any malformation during this intricate process allows for an increased susceptibility to neurodegeneration. The foundation of the CNS occurs during embryogenesis, where approximately 40% of the developing genes primarily regulate CNS development [1]. This starts with the development of the neural plate, which continues to grow and fold onto itself until this groove forms into the neural tube [2]. This developmental precursor relies on specific signals to induce rudimentary CNS formation [2]. At the final stages of development, the CNS can be divided into two sections—the brain and the spinal cord—with both involved in receiving and processing sensory information to induce a biological response [3]. While both the brain and spinal cord are encapsulated within bone, the meninges and cerebrospinal fluid (CSF) act as physical barriers with biochemical protective factors to prevent degeneration [3]. In addition to the meninges and CSF, signaling proteins are essential to the regulation of neuroprotective pathways, simultaneously providing a system of checks and balances to protect the CNS from neurodegeneration. For example, autophagic regulating proteins and ubiquitin were found to have altered expression levels in the CSF of Alzheimer’s disease (AD) and Parkinson’s disease (PD) patients [4,5,6,7].

Consequential neuronal death due to varying biological irregularities lead to the onset of various NDDs such as amyotrophic lateral sclerosis (ALS), frontotemporal dementia (FTD), AD, PD, and Huntington’s disease (HD) [8]. Both AD and PD pathology are caused by the accumulation of protein aggregates resulting from the dysfunction in varying cellular pathways. In AD pathology, it is suggested that an accumulation of autophagic vacuoles are present in the cortex due to an inability to perform or an inhibition of cholinergic neuron mitophagy [4,9,10]. In contrast, PD involves the hallmark histopathological Lewy body detection in mitochondrial-based proteins associated with degraded dopaminergic neurons [4,11,12]. While it is unclear if misfolded proteins contribute to disease development upon aggregation or are merely signs of proper biological processes that gather and eliminate harmful proteins, the etiology is still elusive (Figure 1). It is clear, however, that disrupting neuronal autophagy or progressing presynaptic neuron death results in NDD pathogenesis [8,9,10,11,12,13]. 

While correlations exist for both AD and PD on a genetic and hereditary level, it is still inconclusive whether these diseases are inherited or idiopathic. On the other hand, HD is an autosomal dominant NDD that impacts motor skills and cognition through neuronal loss in the striatum, with an average 15-year life expectancy post-diagnosis [20]. Striatum neuronal degeneration is believed to be triggered by CAG trinucleotide tandem repeats that result in extended polyglutamine sequences in the translated huntingtin protein (Htt). While 6 to 10 polyglutamine sequences in eukaryotic proteins have been identified to facilitate protein-protein interactions (PPIs) through the coiled-coil conjunction within two proteins [21], the CAG trinucleotide repeats within Htt are typically 10–35 units in length; the abnormal gain of function HD-derived mutations allow for upwards of 40–100 repeats that alter Htt localization and accumulation [22,23]. The function of Htt remains experimentally unclear, but its cerebral and basal ganglia localization in the brain suggests its involvement in the regulation of movement coordination. 

During embryogenesis, Htt levels are evenly dispersed throughout progenitor cells that eventually differentiate into striatum-based cortical neurons, whereas in HD fetuses, the Htt protein congregates in the apical end-feet of progenitor cells [24]. This suggests that Htt may play a role in proper CNS development by ensuring proper motor skill development and coordination through the facilitation of progenitor cell polarity and differentiation during neurogenesis. The striatum, being the input model for the basal ganglia, emits signals to dictate learning through action selection and behavior reinforcement [13,25,26]. Classified as a basal ganglia disease, HD phenotypes are typically subtle, with a slow progression, until the patient is unable to move and speak [20]. It can be speculated excess CAG tandem repeats inhibit mutant Htt (mHtt) aggregate clearance and impair essential neurodevelopment-based PPIs due to conformational changes; this is similar to other neurological diseases that lead to aggregate buildup and mitochondrial defects [27]. As observed in HD mice models, developmental pathways that regulate synaptic homeostasis are impaired [28]. Htt has been identified to interact with numerous developmental proteins that regulate 14-3-3 signaling, microtubule-based transport, and proteostasis [29]. A structural analysis of mHtt in any of these essential developmental proteins can determine if the excess CAG repeats truly alter these PPIs, or even whether downstream activation of other pathways is also impaired. 

## 2. Protecting and Maintaining Homeostasis within the Central Nervous System

Each cell type has pertinent roles in maintaining CNS homeostasis. This includes regulating the transfer of ions and molecules across membranes, interpreting and sending signals across neuronal circuits for processing, and mediating the oxidative stress response [30,31]. For instance, microglia act as the CNS phagocytic cells and make up approximately 10% of all CNS cells in the brain [32]. Microglial activation is critical for neuroprotection as it acts as the first line of neuroimmune defense and the downstream induction of an inflammatory response [33]. Upon stimulation by microglial activation, CNS-localized astrocytes and oligodendrocytes [34,35] aid in amplifying conduction and maintaining proper ion and neurotransmitter concentrations [32,34,36]. Astrocytes also compose the protective layer that inhibits the promiscuous entry of substances through the brain’s blood supply. 

The microglial-induced transformation of astrocytes has comprehensive CNS-regulating capabilities that allow for either the promotion of tissue repair or the stimulation of neuronal death upon injury. The facilitation of these intricate processes is dictated upon the subclass of astrocytes activated—the reactive astrocyte (A1) and the neuroprotective astrocyte (A2). It can be implied that A1 astrocytes have an overarching neuroprotective role by stimulating neuronal cell death, despite inhibiting CNS tissue repair that A2 astrocytes typically regulate. Following axotomy, microglial cells secrete the cytokines interleukin-1 alpha (IL-1α), tumor necrosis factor (TNFα), and complement component 1q (C1q) for A1 astrocyte activation [37]. A1 astrocytes have been identified in various NDDs, as improper A1 regulation leads to the natural abundance of neurotoxins that induce rapid neuronal death. 

With regard to the brain, the delicate nervous tissue remains highly vulnerable to biological damage. Due to the brain’s significant role in memory and sensory interpretation, the blood–brain barrier (BBB) acts as a protective layer to facilitate cellular and molecular transport to and from peripheral tissues [35]. Neighboring astrocytes facilitate both partial formation and crosstalk with endothelial cells for peripheral-CNS communication, further solidifying their role in neuroprotection [38,39]. Supplemental astrocyte-driven support allows for the BBB to act as the CNS filter, but it cannot perform independently to maintain neuronal homeostasis [40]. 

Primary detrimental impacts on the BBB are typically due to improper blood supply and flow. Mitigation from ischemic events involves the circle of Willis (CW), an essential component of the midline blood supply to the brain due to irregular artery function, as well as protection from varying arterial pressure changes that can induce susceptibility to physical damage in the BBB [41]. Abnormalities within the BBB and CW have recently been reported in specific AD cases such as increased atherosclerosis in CW and the accumulation of tau and amyloid beta (Aβ) from BBB-induced neuroinflammation [42,43]. While further investigation into the specific pathologies of each NDD is critical for proper diagnosis and treatment, the onset of these irregularities can be traced back to abnormal post-translational modifications, including ubiquitylation. 

## 3. Neuroprotection through Redox Chemistry

From development to the senescent stages of the CNS, metabolically driven processes lead towards excess electrophilic byproducts; thus, alleviating this accumulation is essential for biological homeostasis. Metabolically driven processes in the brain required for proper function account for approximately 20% of the overall energy consumption in the body [44], and these reactions require a properly regulated mitochondrial electron transfer to produce sufficient amounts of ATP. As electrons are passed through the different complexes in the electron transport chain (ETC), both H^+^ and H_2_O are produced from O_2_ reduction by cytochrome c oxidase [45]. The inability to properly reduce O_2_ can cause the production of damaging reactive oxygen species (ROS).

Redox homeostasis helps to the maintain the balance between the production of antioxidants, ROS, and some reactive nitrogen species (RNS) [46]. Many ROS, which includes hydrogen peroxide (H_2_O_2_), hydroxyl radical (^•^OH), hydroperoxyl radical (HO_2_^•^), peroxyl radical (ROO^•^), superoxide (O_2_^•−^), and singlet oxygen (1O_2_), are naturally occurring byproducts of mitochondrial metabolism (Figure 2) [47]. 

Despite this regulatory role, unregulated ROS accumulation can trigger oxidative stress and an increased nucleophilic attack of exposed residues in proteins that are susceptible to ROS/RNS modifications [47,48]. These ROS/RNS-induced PTMs can cause different changes in a protein’s fold and/or function, including increased insolubility and subsequent aggregation due to the exposure of hydrophobic residues in the modified protein [49]. Cysteine residues are particularly susceptible to ROS/RNS modification due to the inherent hyperreactivity of the R-group thiol [50]. If oxidative stress is prolonged and aggregated proteins continue to accumulate, neuronal dysfunction and death can occur, which is a hallmark in numerous NDD. To better understand how to protect against neurodegeneration, it is critical to understand both the molecular effects that ROS has on cellular processes and the underlying mechanisms for how redox homeostasis regulators are activated. These unregulated ROS/RNS PTMs can be mitigated through ubiquitylation by activating the redox homeostasis response, thus leading to the degradation and eventual clearance of protein aggregates.

Protein modifications that are enhanced by RNS accumulation have detrimental impacts on neuroprotection. For example, RNS accumulation after a traumatic brain injury (TBI) is speculated to induce the S-nitrosylation of glyceraldehyde 3-phosphate dehydrogenase (GAPDH). With glyceraldehyde 3-phosphate dehydrogenase (GAPDH), RNS accumulation following traumatic brain injury (TBI) is speculated to influence S-nitrosylation. Through GADPH S-nitrosylation, Sirtuin1 deacetylation is prevented, which then activates p300/CBP acetyltransferase [51]. This leads to TBI-induced tau acetylation at Lys274 and Lys281. The modification of tau at these lysine residues is also observed in AD [52], further suggesting the essential role of regulating RNS-induced PTMs to alleviate both aging and non-aging neurodegeneration. It would therefore be imperative to assess the potential of targeting S-nitrosylated GAPDH for proteolysis as a therapeutic approach for alleviating tauopathy memory loss, AD, or other tau-related NDDs. 

## 4. Interplay between the Ubiquitin-Proteasome System and ROS Production

The ubiquitin-proteasome system (UPS) is a highly regulated mechanism for protein degradation that regulates many biological processes to maintain cellular homeostasis [60]. A protein is targeted for degradation upon ubiquitylation, where the small 8.6 kDa protein, ubiquitin, is covalently attached to the target protein through an isopeptide bond [61]. Ubiquitylation involves the sequential transfer of ubiquitin through a three-enzyme cascade—an ubiquitin-activating enzyme (E1), an ubiquitin-conjugating enzyme (E2), and an ubiquitin ligase (E3). The E2/E3 combination ultimately determines the specific site of attachment of ubiquitin on the target protein and the lysine linkage between ubiquitin molecules within the polyubiquitin chain that is built [61,62,63,64]. The process is initiated when ubiquitin and E1 undergo an ATP-dependent reaction in which a thioester bond is formed between the C-terminus of ubiquitin and the catalytic cysteine on the E1 (E1~ubiquitin) [60]. The activated ubiquitin is then transferred through a transthiolation reaction to a cysteine residue on an E2 enzyme (E2~ubiquitin) [64]. An E3 ligase then facilitates or directly catalyzes the transfer of ubiquitin from the E2~ubiquitin complex to a lysine residue of the substrate to form a stable isopeptide bond between the C-terminus of ubiquitin and the epsilon-amine of the lysine residue on the substrate [64,65]. The type of mechanism used to attach ubiquitin onto a target protein is dependent on the type of E3 ligase—Really Interesting New Gene (RING), Homologous to E6-AP Carboxyl Terminus (HECT), or RING-Between-RING (RBR) [60]. A RING E3 ubiquitin ligase acts as a scaffold to mediate the transfer of ubiquitin from the E2~ubiquitin complex directly to a lysine residue on the substrate [61]. In contrast, a HECT E3 ubiquitin ligase removes ubiquitin from E2~ubiquitin to form an E3~ubiquitin thioester intermediate, which then catalyzes the ubiquitin transfer on to the substrate [60]. A RBR E3 ubiquitin ligase recruits an E2~ubiquitin complex through its RING1 domain, like a RING E3 ligase, but instead of coordinating the transfer of ubiquitin from E2~ubiquitin directly to a substrate, the ubiquitin is transferred to a conserved catalytic cysteine in the RING2 domain (also known as “required for catalysis” (Rcat)) prior to discharge on to a substrate, analogous to the HECT E3 ubiquitin ligase mechanism [66]. 

Ubiquitin can also be removed from a substrate by a deubiquitylase (DUB), which can catalyze the specific hydrolysis of the ubiquitin-substrate isopeptide bond [67]. Recent studies have shown that oxidative stress induces ubiquitylation for aggregate clearance [68], and can influence stem cell differentiation during embryogenesis [69] (Table 1). Maintaining ROS balance through this system may provide a pharmacological approach in neurodegeneration treatment and prevention. Given the advent of recent technologies such as proteolysis-targeting chimaera (PROTAC) [70], molecular glues [71], and the DUB inhibitors in neurological disorders [72], the identification of pertinent enzymes and substrates regulated by ubiquitylation in response to oxidative stress can help to clarify the molecular basis for various NDDs. 

### 4.1. Mediating ROS Production through Transcriptional Activation of Detoxification Genes

The harmonious balance between ROS-producing and ROS-eliminating protein activation is essential for balancing redox homeostasis. For instance, the ROS-producing GTPase Ras-related C3 botulinum toxin substrate 1 (Rac1) is a dedicated neuroprotective protein that is ubiquitously expressed, mediating neurodevelopment and neural health through effector protein binding in its active form [73,74,75,76]. Previous studies have shown Rac1 knockdown in *Drosophila melanogaster* alters the AD-based protein domain network (PDN) interactions that triggers premature astrocytic glial death [77]. Rac1 deficiency is also linked to neural progenitor reduction and primary microcephaly, accompanied by improper striatum and cerebral cortex sizes [77,78]. Missense mutations have also been identified in spastic paraplegia and psychomotor retardation with or without seizures (SPPRS)-like phenotypes such as improper skull size and eye development, intellectual disability, and hypotonia [79,80]. The presence of a RhoGAP domain within Htt suggests that its role may be to activate Rac1 specifically for actin remodeling [81,82]. Further evidence shows an increase in Rac1 activity in the HD striatum tissue, suggesting mutant Htt (mHtt) may not have the capabilities of binding with Rac1 to mediate growth factor signaling [82]. Unfortunately, active Rac1-mediated actin filament organization and polymerization also further increase ROS production [73]. Direct Rac1 and mHtt targeting remains unlikely for HD treatment; this, however, does not detract from the possibility of targeting of proteins indirectly involved in ubiquitylation to regulate either expression and/or Rac1 activation levels. 

The HECT domain and ankyrin repeat containing E3 ubiquitin protein ligase 1 (HACE1) mediates Rac1 degradation through ubiquitylation in a myriad of processes and signals [79,85,86,102,103,104,105,106,107]. For example, the overexpression of HACE1 has been shown to diminish mHtt-induced free radical production through nuclear factor erythroid 2-related factor 2 (Nrf2; aka NFE2L2) antioxidation activation [87], thereby eliminating ROS and maintaining astrocyte mitochondrial function. Due to increased astrocytic activity required for BBB-mediated transport of essential ions and molecules for the brain, whether this be direct chemical signals or nutrients, it is essential to maintain astrocyte-based homeostasis. HACE1 ubiquitylation activates ROS elimination, signaling or targeting substrates for cellular localization, and proteolysis ensures proper astrocyte function. More specifically, when active Rac1 binds to the NADPH oxidase (NOX) complex, HACE1 can recognize, bind, and polyubiquitylate Rac1 to target it for proteasomal degradation [85]. 

The Nrf2 transcription activation system is also tightly regulated through ubiquitylation via Kelch-like ECH-associated protein 1 (Keap1). Keap1 is a RING E3 ubiquitin ligase that utilizes PTMs to form a disulfide bond between two of its cysteine residues, causing a conformational change that inhibits Nrf2 ubiquitylation (Figure 3). This in turn inhibits Nrf2 degradation via the 26S proteasome and allows for Nrf2 nuclear localization upon excess H_2_O_2_ accumulation from mitochondrial metabolism [108]. 

What makes this mechanism of Nrf2 regulation interesting is the requirement of the Cullin-3/Keap1/Rbx1 complex, further supporting the multifunctional roles these proteins play in neuroprotection beyond direct protein–protein interactions for Nrf2 signaling pathway activation or inhibition. Disrupting or even enhancing necessary large protein complexes can serve as a therapeutic approach towards NDD treatment; this, however, requires a full understanding, on a biophysical and biochemical level, of how these complexes form exactly and which residues of each protein are essential to solidify various bonds and interactions.

### 4.2. Mediating Oxidative Stress-Induced Cell Death via Ubiquitylation

Mediating apoptosis induced by oxidative stress is crucial for neuroprotection to mitigate further ROS production. For example, tumor necrosis factor receptor-associated factor 6 (TRAF6) was recently found to activate apoptosis signal-regulating kinase 1 (ASK1) upon excess H_2_O_2_ production to induce cell death [84]. TRAF6 has also been linked to the toll-like receptor 4 (TLR4) signaling pathway through its interaction and activation of nuclear factor-kappa B (NF-κB), thereby participating in the inflammatory response after an ischemic stroke [111]. 

Nitric oxide (NO) accumulation also activates a cascading signal to induce apoptosis in the event that NO accumulation cannot be properly regulated. Brain-derived neurotrophic factor (BDNF) induces NO production and increases cAMP response element-binding protein (CREB) binding activity, a critical regulator of gene expression in dopaminergic neurons [112]. With CREB activation, GAPDH undergoes S-nitrosylation at Cys150, which abolishes GAPDH activity [95]. This irregular PTM leaves GAPDH susceptible to unstable absentia homolog 1 (SIAH1) binding, which in turn leads to SIAH1-dependent ubiquitylation of GAPDH to induce apoptosis [96,113]. SIAH1 can also activate dopamine release through the inhibition of synphilin-1, the primary neurotransmitter depleted in PD patients [96]. However, synphilin-1 is arguably also neuroprotective since it inhibits cytochrome c translocation, MPP+ based ROS production, and apoptosis in PD-impacted dopaminergic neurons [114]. Regardless of potential neuroprotective capabilities, elevated protein levels can disrupt cellular homeostasis and must be tightly regulated through ubiquitylation to control cell death. 

Oxidative stress causes the dysregulation of antioxidant pathways, and those that influence apoptosis are arguably the most critical. However, these antioxidant pathways are still poorly understood. The nuclear translocation of phosphatase and the tensin homolog deleted from chromosome 10 (PTEN) by N-methyl-d-aspartate (NMDA) stimulation has recently been identified to encourage excitotoxicity and ischemic neuronal injury and death [115]. While this modification is known to be mediated through PTEN monoubiquitylation at Lys13 in order to encourage nuclear localization [116], the mechanism to then polyubiquitylate this modified PTEN for cytoplasmic localization and proteolysis is poorly understood in neuroprotection. While WW domain-containing protein 2 (WWP2) has been identified as a HECT E3 ubiquitin ligase that mediates PTEN ubiquitylation [117,118] to induce apoptosis for tumor-cell survival, further investigation into the potential neuroprotective role that WWP2 may play regarding PTEN degradation is needed. For example, influencing PTEN levels in a WWP2-dependent manner in order to alter PTEN cellular localization may provide the therapeutic potential for encouraging neuroprotection following neuronal injury and death. This, however, would be dependent on a better understanding of the potential neuroprotective or toxic role PTEN has, specifically the downstream signaling influenced by NMDA stimulation. 

### 4.3. Neuroprotection through Aggregate Accumulation, Clearance, and Degradation

Neuroprotection can also be fulfilled with fibrous or aggregate clearance through proteolysis, autophagic, or lysosomal pathways. A myriad of signaling pathways are activated through ubiquitylation, including the classic proteolysis of a specific substrate through the 26S proteasome [60]. Various mechanisms that drive this protective function are primarily facilitated through ubiquitylation to cause their colocalization to areas of protein aggregation within the cell, which is not their typical function. It can be speculated that various proteins involved in ubiquitylation may have adapted multifunctional capabilities in order to further provide neuroprotective properties. A prime example is TRAF6, which was found to colocalize in Lewy bodies to facilitate the cytoplasmic aggregation of both Parkinson disease protein 7 (DJ-1/PARK7) and α-synuclein caused by atypical ubiquitylation via K6-, K27-, and K29-ubiquitin linkages [83]. While ubiquitylation for aggregate accumulation are typically caused K63-ubiquitin linkages, both K6- and K29-linkages retain a relatively linear chain formation but a more compact fold, possibly to encourage aggregate clearance and potentially prevent their untimely deubiquitylation. 

DJ-1 is a ubiquitously expressed, redox-sensitive, dimeric chaperone protein that is involved in dopamine synthesis, aggregated α-synuclein clearance, dopaminergic mitochondrial regulation, and oxidative stress-induced transcription regulation [119,120,121,122]. DJ-1 contains three exposed cysteine residues that are susceptible to oxidation, although the highly conserved Cys106 residue has been shown to have the most detrimental effect in DJ-1 activity [123] (Figure 4). 

The classic Lys166Pro mutation in DJ-1 is prevalent in autosomal recessive, early-onset forms of PD [124], likely due to hyper SUMOylation susceptibility and the inhibition of DJ-1 dimerization, although SUMOylation at Lys130 is essential for proper function [124,125,126]. When active, DJ-1 can also eliminate excess H_2_O_2_ generated during mitochondrial metabolism, specifically working to protect mitochondrial complex I [119,125].

Another regulator of α-synuclein ubiquitylation and clearance is ubiquitin-specific peptidase 9 X-linked (Usp9x). Usp9x deubiquitylates monoubiquitylated α-synuclein, which is speculated to trigger α-synuclein clearance through autophagy pathways [94]. This in turn provides an alternative pathway to degrade α-synuclein in lieu of the ubiquitin-proteasome pathway [127]. Preliminary studies reveal that Usp9x may interact with ITCH (a HECT E3 ubiquitin ligase), as increased Usp9x expression levels have been correlated with ITCH deubiquitylation [128]. Further studies would need to confirm this interaction, and, if verified, could also serve as another therapeutic approach for NDD treatment by regulating both ROS and/or aggregated protein clearance. 

## 5. NDD Regulation by ROS and the UPS: Moving Forward

Ubiquitylation plays a critical role in neuroprotection, far beyond its proteolytic degradation targeting activity. This modification is essential in neurodevelopment, neuronal homeostasis mediation, and the prevention of aging-related neurodegeneration. As discussed, this form of a PTM not only targets proteins for degradation, but also utilizes ubiquitin attachment to trigger conformational changes, activates aggregate clearance pathways such as autophagy, maintains CNS glial cell function through its support of complex formations, and enhances protein nuclear trafficking for transcription activation. Various pathways also rely on the properly timed modifications of substrates to regulate the intracellular accumulation of reactive species in preventing neurodegeneration. However, reactive species are also imperative for cell differentiation, cell–cell signaling, the proper timing for the activation of such pathways, and maintaining proper neurotransmitter production. ROS and RNS levels need to be properly regulated since total removal of ROS and RNS would inhibit some of the essential cellular mechanisms that we have discussed in this review. Maintaining ROS levels is essential for homeostasis. Although the current research focus on NDD pathology remains centered on improper protein folding, these recent findings that show the intricate cross-communication between redox chemistry and ubiquitylation may reveal novel neuroprotective functions that could serve as new future NDD drug targets. 

## Figures and Tables

**Figure 1 cells-10-02121-f001:**
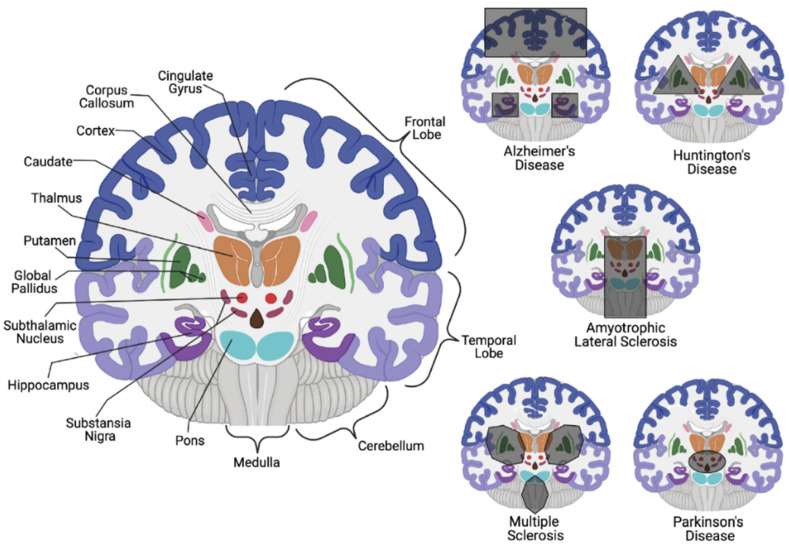
Neurodegenerative disease localization in relation to areas of the brain and the central nervous system. On the left, key parts of the brain are labeled. On the right, regions of the brain that have been identified to be affected by specific neurodegenerative diseases are highlighted. AD has been identified to affect the hippocampus, cingulate gyrus, and many cortical regions [14]. HD affects the caudate, cerebral cortex, putamen, and global pallidus [15]. ALS affects the medulla oblongata, pons, corpus collosum, and thalamus [16]. MS affects the putamen, global pallidus, caudate, thalamus, and medulla [17,18]. PD affects the substantia nigra in the mid-brain most significantly [19]. Figure created with BioRender.com.

**Figure 2 cells-10-02121-f002:**
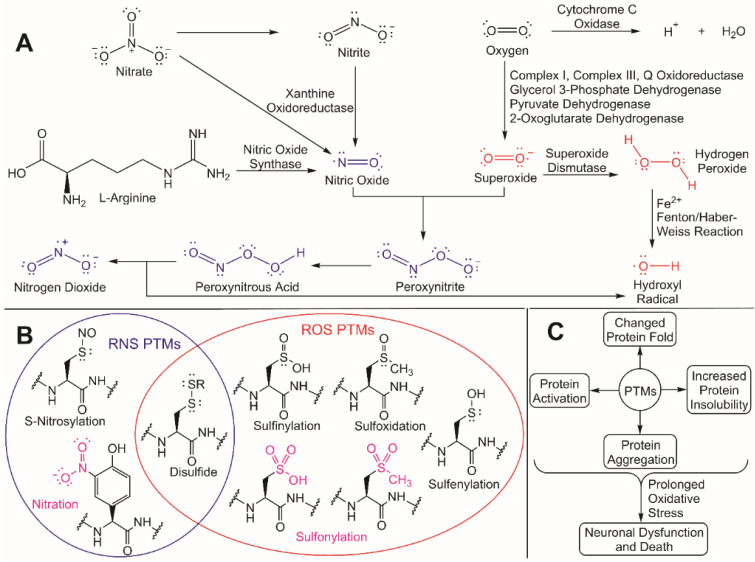
Production of reactive species and corresponding PTMs induced by oxidative stress. (**A**) RNS (blue) and ROS (red) are produced by various enzymes and reactions. Nitric oxide is produced from nitrate and nitrite (catalyzed by xanthine oxidoreductase) and L-arginine (catalyzed by the nitric oxide synthases) [53,54,55]. In the mitochondria, oxygen is reduced to H^+^ and H_2_O by cytochrome c oxidase to promote the production of ATP in the electron transport chain [53]. Superoxide is generated as a byproduct in the mitochondria via complex I, complex II, Q oxidoreductase, glycerol 3-phosphate dehydrogenase, pyruvate dehydrogenase, and 2-oxoglutarate dehydrogenase [53,54]. Superoxide is then either catalyzed by superoxide dismutase to form H_2_O_2_ or reacts with nitric oxide to produce peroxynitrite [53,56]. H_2_O_2_ will undergo an iron-dependent Fenton reaction (also referred to as a Haber–Weiss reaction) to produce a hydroxyl radical [57,58,59], and peroxynitrite will be reduced to generate peroxynitrous acid and can decompose to produce nitrogen dioxide and a hydroxyl radical [59]. (**B**) Reversible (black) and irreversible (magenta) post-translational modifications that are caused by reactive nitrogen species (within the blue circle) and reactive oxygen species (within the red oval). Reversible modifications of cysteine residues caused by reactive nitrogen species include S-nitrosylation and disulfide bond formation, while an irreversible modification caused by reactive nitrogen species includes tyrosine nitration [56]. Reversible modifications caused by reactive oxygen species include disulfide bond formation (cysteine), sulfinylation (cysteine), sulfoxidation (methionine), and sulfenylation (cysteine). Irreversible modifications caused by ROS include sulfonation of cysteine and methionine residues [56]. (**C**) The PTMs generated by ROS and RNS can activate proteins, alter the 3D fold of a protein, increase protein insolubility, and/or cause protein aggregation [54,56]. When oxidative stress is prolonged, these physiological effects can lead to neuronal dysfunction and death.

**Figure 3 cells-10-02121-f003:**
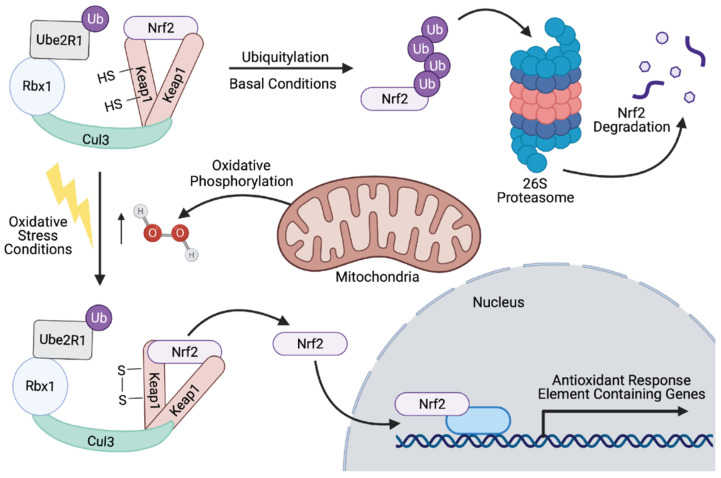
Nrf2 degradation and activation by the Keap1-Cul3-Rbx1 E3 ubiquitin ligase complex. Under basal conditions, the Keap1-Cul3-Rbx1 E3 ubiquitin ligase complex interacts with Ubiquitin Conjugating Enzyme 2 R1 (UBE2R1; E2) to polyubiquitylate Nrf2 [109]. Polyubiquitylated Nrf2 is targeted for 26S proteasomal degradation [97,98,110]. Under oxidative stress conditions generated by increased hydrogen peroxide production from the mitochondria by oxidative phosphorylation, Keap1 forms a disulfide bond between two cysteines, which in turn inhibits Nrf2 polyubiquitylation [108]. Free Nrf2 is then released and migrates to the nucleus where it promotes the transcription of genes containing antioxidant response elements [108]. Figure created with BioRender.com.

**Figure 4 cells-10-02121-f004:**
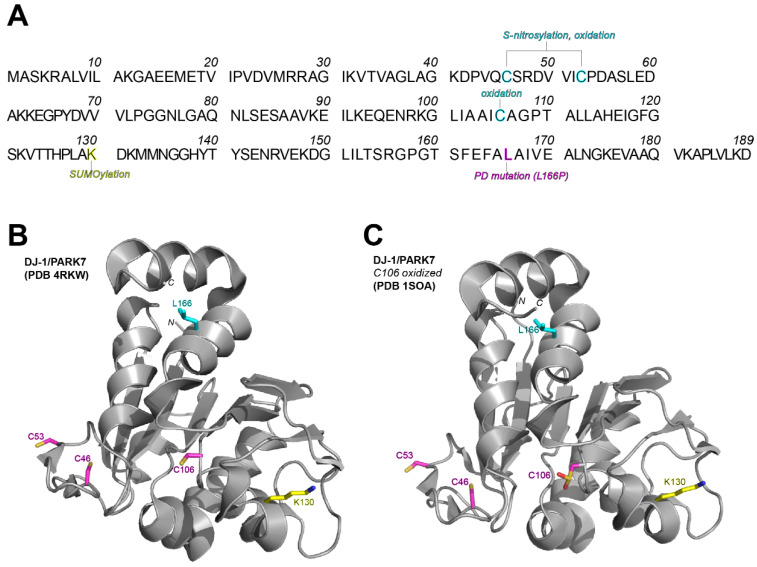
Identification of exposed cysteine residues of DJ-1 and other PTMs necessary for DJ-1 dimerization and activation. (**A**) Primary sequence of DJ-1 (Uniprot Q99497). DJ-1 exposed cysteine residues susceptible to oxidation are highlighted in cyan, the Lys130 residue critical for SUMOylation highlighted in yellow, and the classic Leu166 residue mutated to Pro166 identified in PD is highlighted in magenta. (**B**) The exposed cysteine residues indicate side chain accessibility, and mutations within the structure of DJ-1 can greatly impact the side chain projection in DJ-1 activation. (**C**) In its most active form, DJ-1 is oxidized at Cys 106 (sulfinylation), and an early termination of the α-helix Leu166 residues is predicted to reorient Cys106 and Lys130, which permanently inhibits DJ-1 modification and activation.

**Table 1 cells-10-02121-t001:** Highlighted neuroprotective roles and disease relevance of the ubiquitylation cellular machinery.

Domains and Residue Boundaries	Role in Ubiquitylation	Experimentally Identified Interactor(s)	Neuroprotective Role	Proteopathic Outcome
TRAF6 (Q9Y4K3)				
RING (70–190) TRAF-1 (150–202) TRAF-2 (203–259)	RING E3 Ligase	DJ-1 [83] α-synuclein [83] ASK1 [84]	Redox homeostasis Aggregation clearance Protein trafficking	PD, AD
HACE1 (Q8IYU2)				
AR (62–257) HECT (574–909)	HECT E3 Ligase	Rac1 [85,86] Nrf2 [86] Htt [87]	Redox homeostasis	PD, ALS, HD
ITCH (Q96J02)				
C2 (1–115) WW1 (326–259) WW2 (358–391) WW3 (438–471) WW4 (478–511) HECT (569–903)	HECT E3 Ligase	TXNIP [88]	Redox homeostasis	AD
NEDD4 (P46934)				
C2 (10–160) WW1 (610–643) WW2 (767–800) WW3 (840–873) WW4 (892–925) HECT (984–1318)	HECT E3 Ligase	α-synuclein [89] IGF-1Rβ [90] ABCG1 [91]	Redox homeostasis	AD, PD
NED4L (Q96PU5)				
C2 (4–126) WW1 (193–226) WW2 (385–418) WW3 (497–530) WW4 (548–581) HECT (640–974)	HECT E3 Ligase	BEST1 [92,93]	Redox homeostasis (Ca^2+^/Cl^−^ current balance)	AD
Usp9x (Q93008)				
USP (1557–1956)	DUB	α-synuclein [94]	Proper spine development Neuronal chemical signaling Aggregation clearance	ASD, AD, PD
SIAH1 (Q8IUQ4)				
RING (41–76) SIAH-type (93–153)	RING E3 Ligase	GAPDH [95] synphilin-1 [96]	Oxidative stress response	PD
Cullin3 (Q13618)				
KLHL18 interaction (2–41)	RING E3 Ligase	KEAP1 [97] Nrf2 [98] α-synuclein [99]	Oxidative stress response	AD, PD
Parkin (Q60260)				
RING0 (141–225) RING1 (238–293) IBR (313–377) RING2 418–449)	RBR E3 Ligase	PINK1 [100] Notch1 [101]	Oxidative stress response Aggregation clearance	PD

Highlighted proteins essential in neuroprotection through proper development, maintenance, and redox homeostasis regulation. All proteins listed and the specific mechanisms they employ to ensure neuronal homeostasis are discussed in the text. Domain boundaries listed were extracted from Uniprot (Uniprot code listed in parentheses with the corresponding protein), and identified interactors and disease relevance, with supportive evidence through multiple interaction studies, were obtained through recent literature. Acronyms utilized for domains: AR: Ankyrin Repeat; C2: Ca^2+^-binding domain that associates with phospholipids; HECT: Homologous to E6AP Carboxyl Terminus; IBR: In-between RING fingers; KLHL18: Kelch-Like Family Member 18; RING: Really Interesting New Gene; USP: ubiquitin-specific protease; WW: WW/Proline-rich domain.

## Data Availability

All discussed literature and figures are found in the main text of this mini-review.

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
