# Peer review of "Intersection of Redox Chemistry and Ubiquitylation: Post-Translational Modifications Required for Maintaining Cellular Homeostasis and Neuroprotection"

_cells, 2021, doi:10.3390/cells10082121_

Round 1
Reviewer 1 Report
The manuscript highlights the role of oxidative stress in ubiquitylation and its consequences for protein homeostasis and neurodegeneration. The authors focused only on HECT, Ring and RBR E3 Ligases and a few deubiquitinating enzymes and their role in protein turnover in neurodegenerative diseases.
I have a mixed opinion about this review. Clearly, the chapters 3-5 are focused and are well written and contain valuable information. Whereas, "1. Introduction" and "2. Protecting ... " do not contain a concise message and rather read like bits and pieces from a textbook. In addition, they contain flawed statements such as "CNS localized microglial cells include astrocytes and oligodendrocytes ..." (line 98/99). The cited references should contain more seminal papers especially regarding the work on neurodegenerative diseases. A sentence such as "While age-dependent neurodegeneration remains a hallmark in neurodegenerative disease development" (line 29/30) is a circular conclusion without any meaning. I also do not see why aging-related diseases (AD, PD, ALS...) are named neurodevelopmental diseases (Figure 1). Neurodevelopment usually terminates with the process of myelination in young adults.
additional points:
line 48: CSF or CNS?
line153: nonspecific proteins - whats that?
line176: cystine or cysteine?
line231: actin building or actin bundling?
Author Response
The manuscript highlights the role of oxidative stress in ubiquitylation and its consequences for protein homeostasis and neurodegeneration. The authors focused only on HECT, Ring and RBR E3 Ligases and a few deubiquitinating enzymes and their role in protein turnover in neurodegenerative diseases.
I have a mixed opinion about this review. Clearly, the chapters 3-5 are focused and are well written and contain valuable information.
Authors’ response: We thank the reviewer for this compliment.
Whereas, "1. Introduction" and "2. Protecting ... " do not contain a concise message and rather read like bits and pieces from a textbook. In addition, they contain flawed statements such as "CNS localized microglial cells include astrocytes and oligodendrocytes ..." (line 98/99). The cited references should contain more seminal papers especially regarding the work on neurodegenerative diseases. A sentence such as "While age-dependent neurodegeneration remains a hallmark in neurodegenerative disease development" (line 29/30) is a circular conclusion without any meaning.
Authors’ response: We thank the reviewer for this constructive feedback. We have made appropriate changes to the passages that you pointed out in sections 1 and 2 of our manuscript. This includes adding new passages throughout our text, particularly in sections 1 and 2, and moving some passages to improve the flow, clarity, and take-home messages of our review.
I also do not see why aging-related diseases (AD, PD, ALS...) are named neurodevelopmental diseases (Figure 1). Neurodevelopment usually terminates with the process of myelination in young adults.
Authors’ response: We thank the reviewer for this important feedback. We agree and have modified our manuscript to address this.
additional points:
line 48: CSF or CNS?
Authors’ response: CNS. Addressed
line153: nonspecific proteins – what is that?
Authors’ response: We have changed this statement to “that are susceptible to ROS/RNS-modifications”.
line176: cystine or cysteine?
Authors’ response: cysteine. Addressed
line231: actin building or actin bundling?
Authors’ response: We have clarified this statement. It now reads as “that are susceptible to ROS/RNS-modifications”.
Reviewer 2 Report
This review written by an expert of E3 ubiquitin ligase focuses on the mechanism of interplay between reactive oxygen species/reactive nitrogen species and ubiquitination system in regulation of cellular homeostasis and neuronal survival. In the first part of the review, the authors outlined how ROS/RNS modulates the ubiquitination system in cells. In the second part of the manuscript, the authors gave examples illustrating how ubiquitination of specific proteins in response to oxidative stress contributes to cell death. Specifically, they focused on TRAF6/ASK1, GAPDH/SIAH1, DJ1/PARKIN. Even though the review was brief, it does provide readers a good over view of how ROS/RNS and ubiquitination system respond in concert to regulate cellular homeostasis and neuronal survival. However, this manuscript can be further strengthened by the inclusion of two more findings relevant to neuroprotection: (i) how RNS modulates acetylation of tau through S-nitrosylation of GAPDH and (ii) the role of ubiquitination on nuclear translocation PTEN.
My specific comments and relevant articles on both examples are given below:
- The authors should include the recently published findings by Shin et al. (Cell 184, 2715-2732) on the effects of nitrosylation of GAPDH on acetylation of tau in traumatic brain injury.
- Nuclear localization of PTEN is known to be associated to excitotoxic neuronal death. However, the role of nuclear PTEN in governing neuronal survival is controversial. Although Howitt et al. and Zhang, et al. described in separate reports of nuclear translocation of PTEN in neurons undergoing excitotoxic cell death, their results presented conflicting view of the role of nuclear translocation of PTEN in regulation of neuronal survival – one report suggested it contributed to neuronal death while another report suggested it prevented neuronal death (Howitt, et al. J Cell Biol (2012) 196 (1): 29–36; Journal of Neuroscience 1 May 2013, 33 (18) 7997-8008). The authors should include a discussion of their findings in the review. Additionally, a couple more recent publications relevant to the role of nuclear PTEN in neurological disorders and the mechanism of nuclear translocation of PTEN should be discussed (J Cell Biol (2017) 216 (3): 641–656; J. Biol. Chem. (2018) 293(24) 9292–9300).
Minor comment:
- Line 120, Page 3: “Alleviating electrophilic accumulation from varying biological processes…” The authors should define the meaning of electrophilic accumulation. References to this statement should be cited.
Author Response
This review written by an expert of E3 ubiquitin ligase focuses on the mechanism of interplay between reactive oxygen species/reactive nitrogen species and ubiquitination system in regulation of cellular homeostasis and neuronal survival. In the first part of the review, the authors outlined how ROS/RNS modulates the ubiquitination system in cells. In the second part of the manuscript, the authors gave examples illustrating how ubiquitination of specific proteins in response to oxidative stress contributes to cell death. Specifically, they focused on TRAF6/ASK1, GAPDH/SIAH1, DJ1/PARKIN. Even though the review was brief, it does provide readers a good over view of how ROS/RNS and ubiquitination system respond in concert to regulate cellular homeostasis and neuronal survival. However, this manuscript can be further strengthened by the inclusion of two more findings relevant to neuroprotection: (i) how RNS modulates acetylation of tau through S-nitrosylation of GAPDH and (ii) the role of ubiquitination on nuclear translocation PTEN.
Authors’ response: We thank the reviewer for their compliments and helpful feedback.
My specific comments and relevant articles on both examples are given below:
- The authors should include the recently published findings by Shin et al. (Cell 184, 2715-2732) on the effects of nitrosylation of GAPDH on acetylation of tau in traumatic brain injury.
Authors’ response: We thank the reviewer for this constructive feedback. We have incorporated a new passage on nitrosylation of GAPDH and references to section 3 of our manuscript.
- Nuclear localization of PTEN is known to be associated to excitotoxic neuronal death. However, the role of nuclear PTEN in governing neuronal survival is controversial. Although Howitt et al. and Zhang, et al. described in separate reports of nuclear translocation of PTEN in neurons undergoing excitotoxic cell death, their results presented conflicting view of the role of nuclear translocation of PTEN in regulation of neuronal survival – one report suggested it contributed to neuronal death while another report suggested it prevented neuronal death (Howitt, et al. J Cell Biol (2012) 196 (1): 29–36; Journal of Neuroscience 1 May 2013, 33 (18) 7997-8008). The authors should include a discussion of their findings in the review. Additionally, a couple more recent publications relevant to the role of nuclear PTEN in neurological disorders and the mechanism of nuclear translocation of PTEN should be discussed (J Cell Biol (2017) 216 (3): 641–656; J. Biol. Chem. (2018) 293(24) 9292–9300).
Authors’ response: We thank the reviewer for this helpful feedback. We have addressed this comment by adding a new paragraph and references on the role of PTEN in neuron survival to section 4.3 of our manuscript.
Minor comment:
- Line 120, Page 3: “Alleviating electrophilic accumulation from varying biological processes…” The authors should define the meaning of electrophilic accumulation. References to this statement should be cited.
Authors’ response: We have clarified this statement. It now reads as “From development to senescent stages of the CNS, metabolically driven processes lead towards excess electrophilic byproducts, so alleviating this accumulation is essential for biological homeostasis.”
Reviewer 3 Report
The present review is a very good entry in the intricate array of ubiquitylation involvement in cellular homeostasis and neuroprotection.
The manuscript is very well organized, each section is properly described. figures are clear and informative. references are adequate and updated.
I only suggest to cite the following manuscript in the paragraph speaking about microglia and astroglia interconnected role (page 3, add after ref 29)
Villa V, Thellung S, Bajetto A, Gatta E, Robello M, Novelli F, Tasso B, Tonelli M, Florio T. Novel celecoxib analogues inhibit glial production of prostaglandin E2, nitric oxide, and oxygen radicals reverting the neuroinflammatory responses induced by misfolded prion protein fragment 90-231 or lipopolysaccharide. Pharmacol Res. 2016, 113, 500-514. doi: 10.1016/j.phrs.2016.09.010.
I suggest
Author Response
The present review is a very good entry in the intricate array of ubiquitylation involvement in cellular homeostasis and neuroprotection.
The manuscript is very well organized, each section is properly described. figures are clear and informative. references are adequate and updated.
Authors’ response: We thank the reviewer for sharing our enthusiasm on this topic.
I only suggest to cite the following manuscript in the paragraph speaking about microglia and astroglia interconnected role (page 3, add after ref 29)
Villa V, Thellung S, Bajetto A, Gatta E, Robello M, Novelli F, Tasso B, Tonelli M, Florio T. Novel celecoxib analogues inhibit glial production of prostaglandin E2, nitric oxide, and oxygen radicals reverting the neuroinflammatory responses induced by misfolded prion protein fragment 90-231 or lipopolysaccharide. Pharmacol Res. 2016, 113, 500-514. doi: 10.1016/j.phrs.2016.09.010.
Authors’ response: We thank the reviewer for this helpful feedback. We have added the suggested reference.
I suggest
Round 2
Reviewer 1 Report
The authors considered the points raised by the reviewer and amended the manuscript.